# Spatial production or sustainable development? An empirical research on the urbanization of less-developed regions based on the case of Hexi Corridor in China

**Huailin Zhang[1,2], Zhibin Zhang [1]\*, Jianhong Dong[1], Fawen Gao[1], Wenbin Zhang[1], Weimin Gong[1]**

1 College of Geography and Environmental Science, Northwest Normal University, Lanzhou, China,
2 School of Economics and Management, Hexi University, Zhangye, China

\* zbzhang@nwnu.edu.cn

## Abstract

The relationships between spatial production, urbanization and sustainable development are becoming a focus of the international academic cycle. Urbanization dominated by spatial production driven by capital and power often produces and enlarges uneven development, which leads to multiple eco-environmental problems. Thus, the key to development lies in whether the pattern of urbanization is in harmony with the ecological environment. However, previous researches mainly concentrate on spatial production in developed countries or regions. The urbanization and sustainable development of less-developed regions, with complex and fragile ecological environments, are often overlooked. It is a new idea to explain the relationships and interactions between spatial production, urbanization and sustainable development based on less-developed regions by the theory of spatial production. The paper chooses the Hexi Corridor as a typical case, puts forward a conceptual framework and explores the process of spatial production from 2000 to 2017. The results reveal that urbanization in the Hexi Corridor is a multidimensional socio-spatial process: power and capital gave birth to a higher urbanization and accelerated the process of urbanization, however, the urban-rural gap between regions has not narrowed accordingly. Driven by comprehensive interests, local governments often take some extreme measures to forcefully promote the urbanization process, thereby violating the goals and requirements of sustainable development. At present, there is an urgent need to coordinate the relationship between urban and rural regions on different scales and transform the urbanization model from traditional spatial production to a new-type of urbanization with people-oriented and sustainable development.

## 1. Introduction

Urbanization is an important issue that geography has long been concerned about [1]. In recent years, the process of the urbanization has produced huge and rapid changes worldwide

**Data Availability Statement:** All relevant data are included in the Supporting Information files.

**Funding:** This work was supported by the National Natural Science Foundation of China (grant no 41961029; http://www.nsfc.gov.cn/) and Philosophy and Social Science Planning Project of Gansu Province (grant no YB14098; http://gansu.gscn.com.cn/). The funder contributed to study design.

**Competing interests:** The authors have declared that no competing interests exist.

[2, 3], and the development of urbanization is having a profound impact on China and the world in the era of economic globalization [4, 5]. Compared with developed countries, developing countries could rely on rapid urbanization to achieve economic development and reduce regional income gaps. At present, the center of gravity of global urbanization has shifted to developing countries [6]. As the largest developing country in the world, China, with its rapid urbanization and unique development path, has attracted widespread and increasing attention [7, 8]. From 1978 to 2017, the urbanization rate of China has increased from 17.9% to 58.52%; the number of people living in poverty has dropped sharply, and public infrastructure, education, medical care and social security have been improved. At the same time, the rapid urbanization process has caused the disorder of spatial production, which has brought a series of spatial contradictions and social problems, and has made sustainable development face serious challenges [9].

Urbanization and the resulting allocation of spatial resources have promoted economic growth, changed land use patterns, reshaped regional development patterns, and affected and changed the natural and social environment in many ways [10], for example, environmental degradation [11], urban housing problems [12], urban-rural inequality [13], the widening rural-urban gap [14–16], excessive land development, arable land loss [17, 18], and increased energy consumption [19]. In turn, this change also exerts an influence on urbanization, which is mainly reflected in the planning that determines regional economic development and urbanization blueprints [20]. The process of urbanization is also an essential process of spatial production. Spatial production has the characteristics of social construction, which means that various elements interact and are always are always in the process of being created [21]. At different spatial scales, the social relations shaped by spatial production are also undergoing new changes. Therefore, new theories are urgently needed to guide and improve the traditional urbanization pattern [22].

Urbanization, spatial production and sustainable development are dynamic, multi-dimensional social spatial processes. Many scholars have already studied the issues of urbanization from the perspective of social space dialectics [23–25]. Urbanization studies have shown an important "spatial turn" [26, 27]. China's urbanization is not copied from developed countries; on the contrary, it has distinct Chinese characteristics [28, 29]. Spatial production driven by power and capital has been the main pattern of urbanization in China. The characteristics of this pattern is the coexistence of high urbanization rate and high growth rate, a large number of rural people entering the city, and the rapid expansion of urban scale. Simultaneously, the sharp decline in arable land, water shortages, degradation of oasis, and intensification of "heat islands" effect have caused widespread concern around the world [30, 31]. Under the interaction of urbanization and spatial production, physical space has been greatly improved, and people's sense of gain and happiness has been greatly satisfied. But the spiritual space has been severely squeezed; the neighborhood is isolated; homesickness is lost, and the sense of belonging has weakened. Spatial production is the result of the struggle between different spaces [32]. In the process of urbanization, factors such as power and capital create new space forms in accordance with their own needs, which in itself is a damage to the original spaces. Prosperity and depression, tolerance and exclusion, and new social relations are driving the transformation of social space.

Spatial production and sustainable development have been hot topics in the field of urbanization research [33–40]. Many scholars have conducted extensive research on urbanization in developed countries [41, 42] or developed regions in developing countries [43, 44], but urbanization and sustainable development in less-developed regions with complex and fragile ecological environments are often overlooked. In order to effectively respond to the sustainable development problems associated with urbanization and spatial production, the Chinese

government has formulated a strict ecological supervision system and vigorously promoted the strategy of ecological civilization. However, because the destructive power of economic activities on natural space far exceeds the resilience of natural systems and the ecological carrying capacity of natural resources, the effect is not significant, and in some regions irreparable environmental disasters are even caused. The unsustainable urbanization development pattern has intensified these spatial contradictions and distorted urban-rural relations [45]. At present, the mechanism of urbanization and spatial production is not clear and needs to be supplemented and improved. Based on researches of urbanization and spatial production by Ye et al. [26, 44], this paper proposes a conceptual framework of "spatial production-urbanization-sustainable development", which helps to formulate policies and meet challenges in less-developed regions, and provides a scientific reference for sustainable urbanization in developing countries.

The section 2 introduces the location characteristics of Hexi Corridor and the reality of urbanization, focusing on the interpretation of the theoretical framework of the content and basic ideas. Section 3 describes the spatio-temporal processes of urbanization and spatial production in Hexi Corridor and their impact on the ecological environment. The fourth part emphasizes the consequences of spatial production driven by power and capital, and puts forward the important practical significance of changing the urbanization pattern to the less-developed regions.

## 2. Materials and methods

### 2.1. Study area

The Hexi Corridor (92˚21′-104˚45′E, 37˚15′-41˚30′N) is located in the arid desert area of Northwest China. The geology and geomorphology are complex and diverse, and the ecological environment is very fragile. The total land area is $27.6 \times 10^4$ km$^2$, including five prefecture-level cities (Wuwei, Zhangye, Jinchang, Jiuquan and Jiayuguan), 20 counties (districts), including 4 minority autonomous counties (Akesai, Subei, Sunan and Tianzhu). Nourished by the three inland river systems (Shiyang River, Hei River and Shule River) originating from the Qilian Mountains, the Hexi Corridor has been a famous oasis agricultural demonstration area and irrigated agricultural area in the northwest inland. The area of arable land is 2 million hm$^2$; the area of desert and desertification land has reached 23.6 million hm$^2$, and water resources are extremely scarce. At the end of 2017, the total population was 4,879,200, and the urbanization rate reached 61.86%, which was 14.17 and 3.34 percentage points higher than the average levels of Gansu and China, respectively. Jiayuguan has the highest urbanization rate at 93.45%, which is slightly lower than that of developed coastal cities such as Shenzhen, while Gulang County has the lowest urbanization rate, only 26.2%. The urbanization development within the Hexi Corridor is imbalanced, and is affected and restricted by social economy and natural conditions [46]. Thus, the traditional urbanization pattern can no longer meet the needs of development, and the urbanization transition is imminent.

### 2.2. Data acquisition

The empirical data of urbanization development are mainly from the Gansu Yearbook (2001–2009) and Gansu Development Yearbook (2010–2018) and the statistical bulletin in counties and districts. There are no districts or counties in the Jiayuguan City, and when using the county as the basic unit of analysis, statistics from Jiayuguan City are used instead. Anxi was renamed Guazhou in 2006, and the data before 2006 were obtained from Anxi. Water resource data are from the annual "Water Resources Bulletin" issued by the Gansu Provincial Department of Water Resources, and the LULC data are taken from the Resource and Environment

Data Cloud Platform (http://www.resdc.cn/) of Chinese Academy of Sciences. The case of ecological environment destruction in the Qilian Mountain Nature Reserve provided good support for the paper.

## 2.3. Methodology

Spatial production is not only the reproduction of social space, but also the reproduction of spatial relationships [26]. Space is the product and content of society, and society is the collection and representation of space. The two are constantly being reshaped under the interweaving action [47]. Space is not just a passive recipient of social, economic and cultural infestation, but an active participant in urbanization [48]. Capital, power, and class are the sources of motivation for spatial production and social revolution. Spatial production is a social relationship under the influence of various forces like capital, and then this social relationship has shaped an urbanization process from rural space to urban space [49]. These processes directly affect the three sustainable development areas of economic prosperity, social development and environmental protection [50], as shown in Fig 1.

In recent years, the spatial production caused by urbanization has been more and more unsustainable, the space contradiction has been increasingly acute, and the space relationship has been more complicated. On the social scale, on the one hand, the urbanization dominated by spatial production has changed the social and cultural landscape and improved people's living standards. On the other hand, it also causes the widening income gap between urban and rural regions, and frequent occurrence of social stratification and spatial injustice [51]. On the natural scale, people pay more attention to the living environment and air quality, and the increasing demands for leisure, vacation and recuperation, which force the urbanization to strengthen the environmental protection and ecological restoration [49, 52]. Nevertheless, the urbanization still inevitably leads to resource overload, energy overexploitation, environmental damage and biodiversity loss. Increasing environmental and ecological erosion has accelerated social-ecological interaction between space and the environment [27, 53]. Therefore, it is an inevitable choice for sustainable development to study the mechanism of spatial production and urbanization, and to explore the urbanization pattern in harmony with ecological environment [54, 55].

The interaction between urbanization and spatial production shows that we must pay attention to the coordinated development of the coupling between human and nature, and conduct comprehensive research from an interdisciplinary perspective [56]. Based on the theories of spatial production, urbanization and sustainable development, the article proposes a new

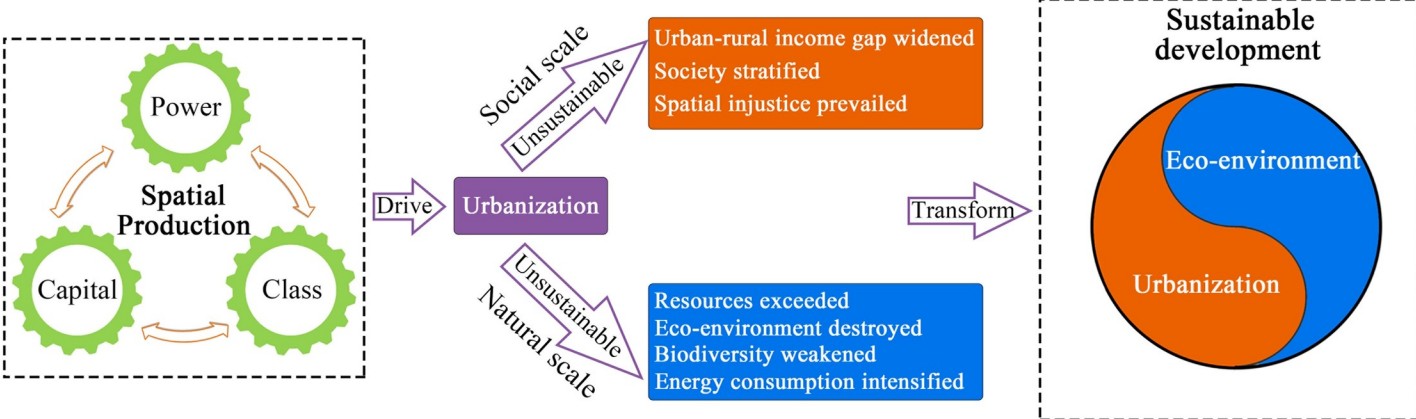

**Fig 1. The conceptual framework of spatial production-urbanization-sustainable development.**

conceptual framework [44, 49, 55, 57, 58], links spatial production, urbanization and the social space and ecological environment problems, and analyzes the process of spatial production and urbanization in less-developed regions. Considering the regional differences and characteristics, we select the variables of FAI (fixed asset investment) and REI (real estate investment), PCFE (per capita fiscal expenditure) and URIR (urban-rural income ratio) to reflect the impact of capital, power and class on urbanization [26, 59], and urbanization rate is used to measure urbanization level. Using socio-economic statistical data, this paper empirically analyzes the evolution characteristics and the spatial pattern of spatial production and urbanization in the Hexi Corridor at the county scale [60]. Urbanization in less-developed regions is at the expense of resources and environment [61], which arouses the world to attach great importance to the sustainable development. This paper describes the changes of resources and environment in the process of urbanization using water resources data and land resources utilization data interpreted by RS (remote sensing) method [62, 63]. The results show that the transformation of urbanization or spatial production mode should not only deal with the relationship among capital, power and class and urbanization, but also actively respond to the economic, social and environmental challenges that sustainable development may face.

## 3. Results: Spatio-temporal evolution of urbanization and spatial production of the Hexi Corridor

The Hexi Corridor is situated at the junction of the Tibetan Plateau and the Inner Mongolia plateau. It is bordered by the Qilian Mountains in the South, and the Badain Jaran Desert and the Tengger Desert in the north. Under the transportation and accumulation of three major regional rivers, the starry Gobi oasis was formed. Known as the "northwest granary" and the ancient Silk Road, Hexi Corridor used to have pleasant climate, developed agriculture and dense population. With a long history and rich cultural resources, Hexi Corridor has played an important role as a bridge and link between Chinese and Western cultures.

At present, the global climate change leads to water shortage, desertification, and oasis area sharp reduction, which seriously affect the ecological environment of the Hexi Corridor. For a long time, development and protection are its unavoidable practical problems. In order to improve the living environment in the western region, China has implemented the policy to develop the western region. The local economic situation has improved significantly; the number of people living in poverty has decreased greatly, and the protection of the natural environment has risen to a national level. The advancement of urbanization strategy has promoted the regional status of the Hexi Corridor to a certain extent, and changed the development pattern of social space. However, the rapid urbanization has caused irreversible damage to the natural environment [64], which has seriously threatened the national ecological security. The changes of geopolitics and the ecological environment restrict the development of this region. For a long time, although the national and local governments have paid more attention to the ecological governance in this region and taken practical actions, the deterioration of the ecological environment has been on the rise. The implementation of the China's Major Function Zoning and ecological civilization have further established the irreplaceable position of the Hexi Corridor at the regional and even national ecological level. It's time to change the way of spatial production in the Hexi Corridor.

### 3.1. Time series analysis of the urbanization evolution in the Hexi Corridor

The rise and fall of the Hexi Corridor is always closely linked to the government's policies of the same period. Since 2000, in addition to the local people's willingness to pursue their own development, the urbanization of the Hexi Corridor has been gradually promoted by the

government. Facts show that power is leading and dominating the process of urbanization in the Hexi Corridor.

The central government documents mainly guide regional development through macro-control. The 10th Five-Year Plan proposes for the first time to build a reasonable urban system and break the urban-rural division system. The 11th Five-Year Plan emphasizes the Longhai Line as the horizontal axis and the urban agglomeration as the main body to form a reasonable spatial pattern of urbanization. The 12th Five-Year Plan proposes to cultivate and expand western urban agglomerations to promote the integrated construction and networked development of urban infrastructure. The requirements of the 13th Five-Year Plan are more specific, emphasizing human's urbanization as the core, narrowing the urban-rural gap, and promoting the urban-rural integration. In the same period, related documents such as the western development, urban system construction, and New-type urbanization planning have been issued in detail to supplement the urban axes and urban belts, urban agglomerations in the western regions, and the Silk Road Economic Belt.

In addition to the "Five-Year Plan" of the central government, in order to actively respond to the overall arrangement of the state, the local government further refined and supplemented the relevant urbanization policies. The policies are mainly based on regional characteristics, and aiming to support and help the Hexi Corridor to achieve the set goals in stages. They put forward the development goals of the Hexi Corridor for the future, that is, to gradually build the Oasis Economic Zone, Hexi Corridor Economic Zone, and Hexi Corridor Urban Agglomeration.

Since 2000, in order to accelerate the urbanization, the central and local governments have continuously increased fiscal expenditures and transfer payments. PCFE has grown rapidly, from 710.25 Yuan (2000) to 12175.33 Yuan (2017), an increase of nearly 940 yuan. PCFE changes can be divided into three stages: low-speed growth (2000–2006), medium-speed growth (2007–2010), and high-speed growth (2011–2017), which are very similar to the change trend of the urbanization rate (see Figs 2 and 3). This change process is also highly consistent with the major policy adjustments of the two levels of government in the same period, further explaining the dominant role of power in the urbanization of the Hexi Corridor.

As the most basic factor of production, capital plays an essential role in the process of urbanization. The urbanization of less-developed regions is quite different from that of other regions. The urban and rural infrastructure is serious insufficiency. The construction intensity is high, and the density is big. To strengthen infrastructure, improve public services and promote the construction of key ecological function areas, we need a large amount of financial support. Since 2000, the central and local governments have continuously increased their policy support and financial transfer payments, FAI and REI have achieved rapid growth, as shown in Figs 2 and 3. FAI amounted to 7.59 million yuan in 2000 and peaked at 264.42 million yuan in 2016, a nearly 35 times. REI also increased 26 times, from 0.14 million yuan in 2000 to 3.77 million yuan in 2016. Although the government had made great achievements in promoting the equalization of public services and narrowing the urban-rural gap, it was difficult to make up the huge historical deficit in the short term. Meanwhile the government investment alone is not a long-term solution to the actual problems due to the imperfect market development in less-developed regions. In 2016, the government reformed the financial investment system and gradually relaxed restrictions on private capital. Private capital has begun to flow into basic public services such as infrastructure, education, and medical care. As a result, Hexi Corridor's FAI and REI began to decline in 2017. Through financial reforms, the healthy development of urbanization has been promoted, and the regional economic and social development has realized the two-wheel drive of state-owned capital and private capital. However, the change in investment structure reflects the role of the government, which further illustrates the impact of power on urbanization.

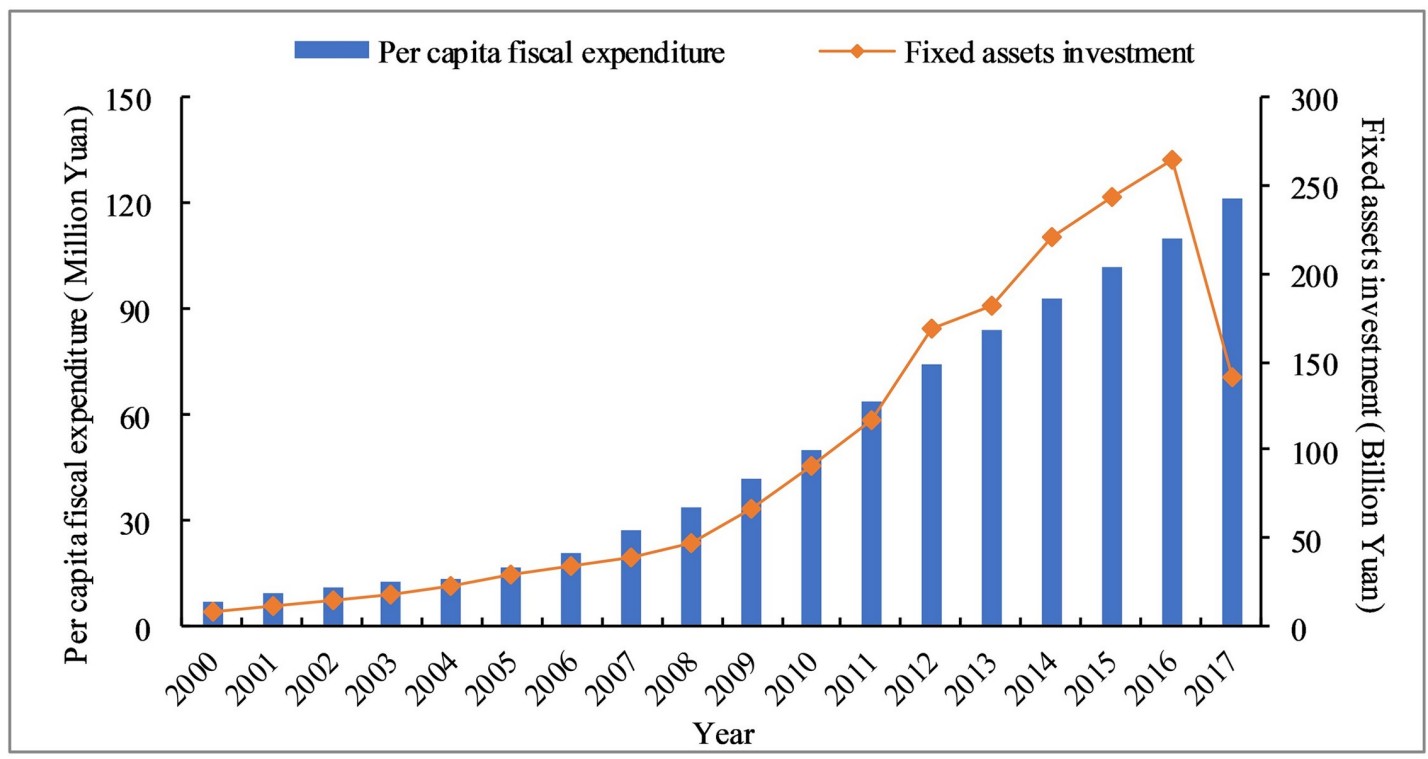

**Fig 2. The variation in PCFE and FAI of Hexi Corridor.**

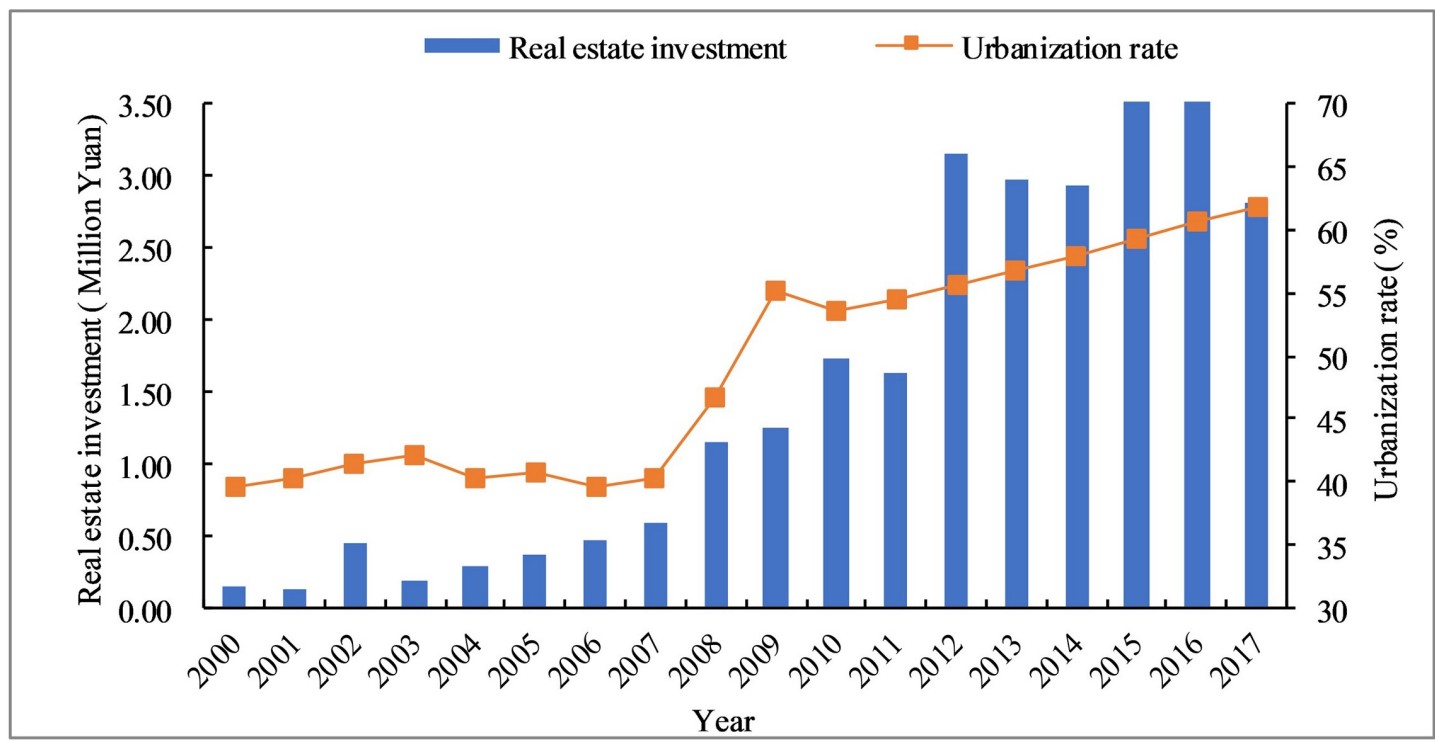

**Fig 3. Evolution of REI and the urbanization rate of Hexi Corridor.**

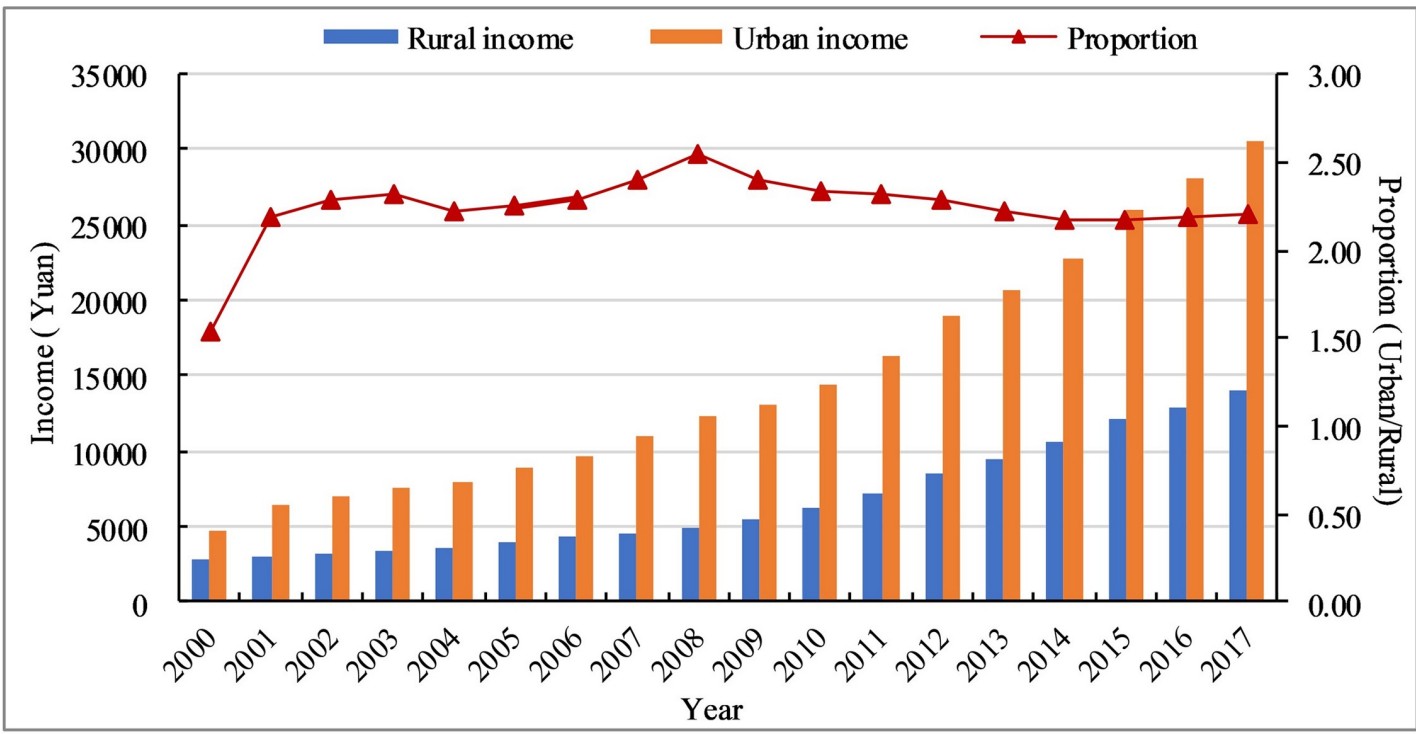

**Fig 4. Change of the urban-rural income gap of the Hexi Corridor.**

The long-standing dual economic structure in less-developed regions is the main cause of urban-rural polarization. Urbanization is an effective way to eliminate polarization and narrow the urban-rural gap. Changes in the structure of agricultural industry have made it possible for surplus labor to enter the cities, and farmers' desire for a better life has accelerated the urbanization. However, the lack of labor skills, barriers to entry for citizenship, and partial coverage of basic social security have left farmers in cities for the first time in an awkward position. The urbanization of the collusion of capital and power cannot maintain the basic income level of farmers. Income imbalance deprives farmers of the benefits and welfare that accrue from being "urbanized", and the excessive income gap deviates from the original intention of "people-oriented" urbanization.

According to the proportion (URIR), we can divide the urbanization process of the Hexi Corridor into two stages (see Fig 4). The period 2000–2008 was the first stage and the URIR continues to increase, reaching a maximum of 2.54 in 2008. Urban and rural income growth has been slow, and the urban-rural incomes gap has continued to widen. In the second stage, the URIR declined slowly, and was basically the same in 2017 as in 2001. Urban and rural incomes have grown rapidly, and the urban-rural income gap has gradually narrowed. Overall, from 2000 to 2008, urban income grew significantly faster than rural income. After 2008, the situation was reversed, and 2008 was a turning point. It is the uneven geography, the unequal distribution of infrastructure and resources, and the differentiated policies that have led to an obvious social differentiation in the Hexi Corridor.

## 3.2. The evolution process of spatial production in the Hexi Corridor

As a result of the rapid urbanization in China, the urbanization rate of the Hexi Corridor continues to increase (see Fig 5), and regional central cities such as Jiayuguan, Suzhou and Jinchuan are generally higher than non-central cities. Although the rate of urbanization growth

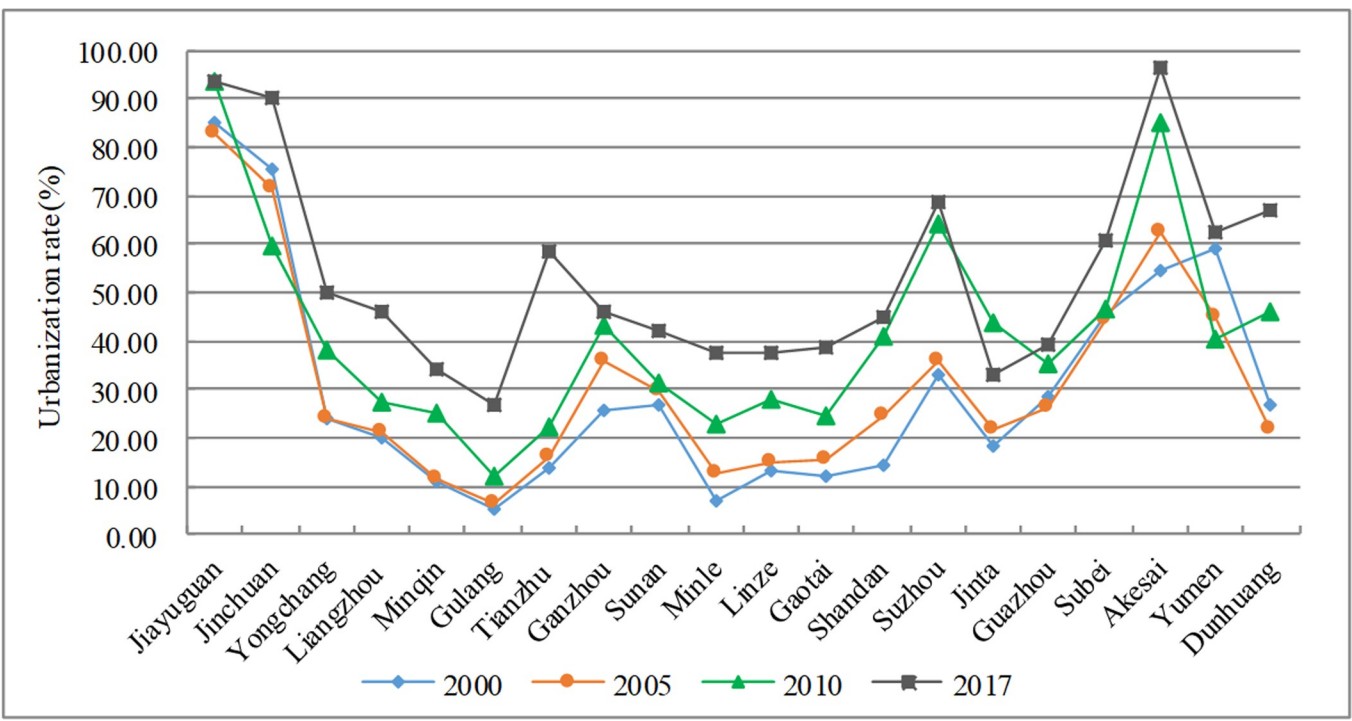

**Fig 5. Evolution of the urbanization rate of the Hexi Corridor.**

varies from city to city, the gap is narrowing, from 79.51 in 2000 to 66.8 in 2017. Some ethnic minority regions have relatively high urbanization rates because of factors like geography, population, and national poverty alleviation policies. The reason is mainly the result of the combined action of FAI, REI, PCFE and so on.

Reducing labor intensity and increasing capital intensity is the only way to activate rural assets and resources and realize rural and agricultural modernization. As a direct driver of urbanization, capital provides a sustainable and stable source of funds for high-quality urbanization. From 2000 to 2017, the amount of FAI in almost every city increased significantly (see Fig 6). Accordingly, the urbanization rate also increased. However, the city where the prefecture-level capital is located had a faster increase in FAI and is much larger than other cities. In addition, the growth of FAI between cities was very uneven, and the gap continued to widen, increasing from the original 1.01 billion yuan to 17.19 billion yuan in 2017, an increase of 17 times. The study found that the factors affecting spatial production varied in the same region. For example, in the Jiuquan region, Suzhou was mainly dependent on FAI to promote the urbanization process, while Akesai's urbanization was mainly subject to population changes. This phenomenon is very common in the minority autonomous counties in the Hexi Corridor, and the higher urbanization rate also indicates that minority policies (i.e., power) have played an important role in the process of spatial production.

In the urbanization with Chinese characteristics, fiscal policy plays an irreplaceable role. Fiscal policy promotes the coordinated development of urban and rural regions, and optimizes the allocation of resources. It is not only the driving force of urbanization, but also an important tool for urban development and rural revitalization. Fiscal expenditure is an important part of fiscal policy and has played an important role in the urbanization of less-developed regions. In 2000, the PCFE of the Hexi Corridor was 35.51 yuan per person, which increased to 608.77 yuan per person in 2017, about 17 times that of 2000. The urbanization rate increased

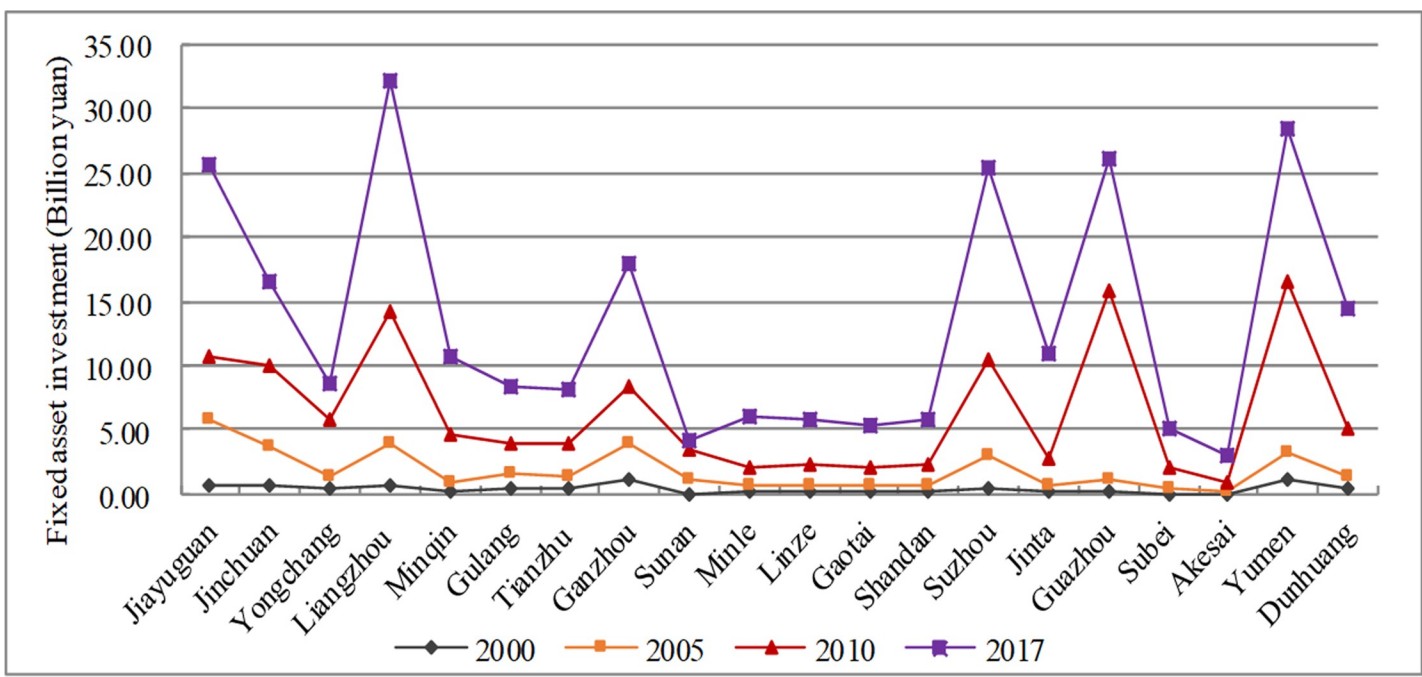

**Fig 6. Evolution of FAI of the Hexi Corridor.**

by 22 percentage points simultaneously. Compared with other regions, minority regions have grown faster (see Fig 7). For example, Tianzhu's PCFE increased from 4.88 yuan per person in 2000 to 211.43 yuan per person in 2017, an increase of nearly 43 times. Compared with other countries, the Chinese government stress people-oriented urbanization and pay more

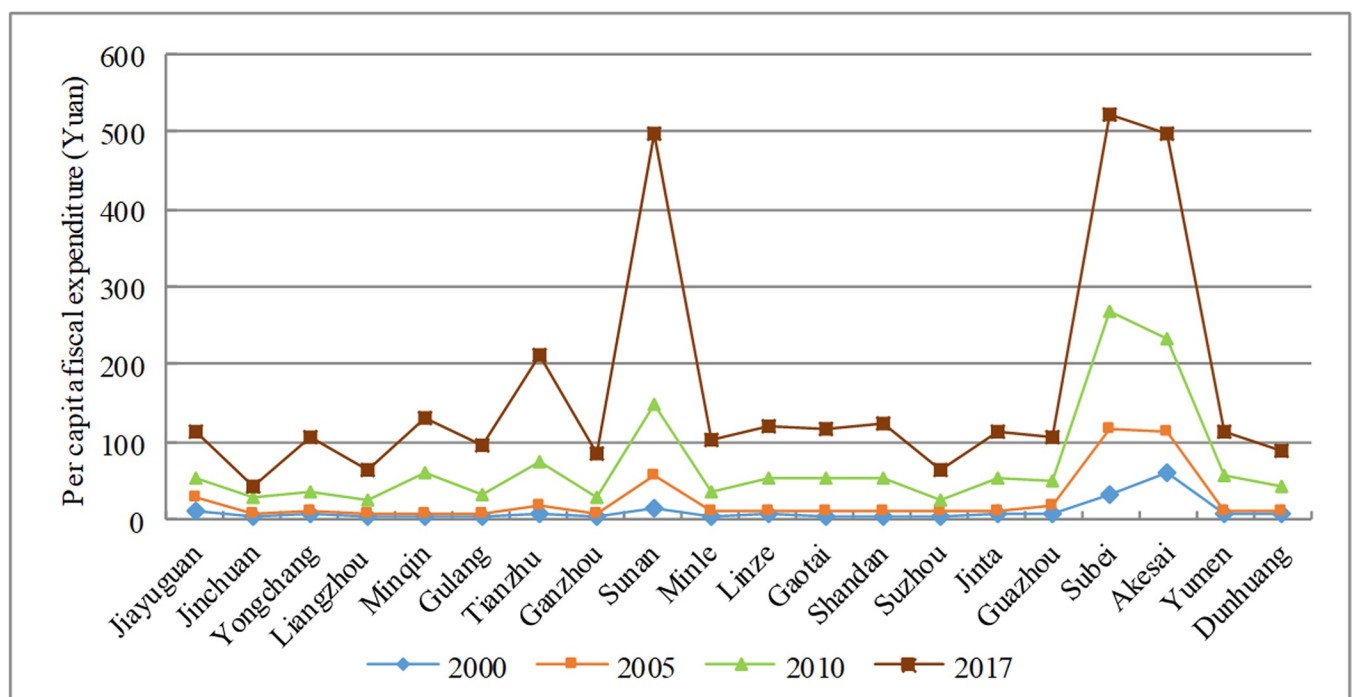

**Fig 7. Evolution of PCFE of the Hexi Corridor.**

attention to improving of people's living standards. Especially through urbanization, the living standards of ethnic minorities have been greatly improved.

Narrowing the urban-rural income gap is not only an important goal of new-type urbanization, but also a basic requirement of the sustainable development. The urban-rural income ratio reflects the differentiation of the rich-poor gap in the social classes and the basic characteristics of the uneven development [44]. In China, urban and rural regions have been two long-standing social classes since ancient times, and they are also the main carriers for the generation and development of social contradictions. Changing the traditional urban-rural dual structure and building a harmonious urban-rural relationship are the only way to achieve sustainable development. Urban and rural regions are two different spaces, and there are huge gaps in public infrastructure, medical services, education, technology, welfare and income [26, 47, 65]. However, spatial production driven by power and capital often ignores the interests of farmers-a phenomenon that has eased after promoting people-oriented urbanization. From 2000 to 2017, the URIR of the Hexi Corridor first rose and then declined (see Fig 8), which means that the urban-rural income gap has gradually narrowed. But within regions, regional imbalances still coexist. In some regions, such as Akesai (1.45), Subei (1.46), and Sunan (1.59), mainly dominated by animal husbandry, with the help of the ecological civilization and the policy of turning marginal arable land to forests, the income and lives of herders have greatly improved. On the contrary, in poor geographical regions, such as Gulang, the urban-rural income ratio is at a high level (3.28), which is higher than the national average (1.5).

### 3.3. Deteriorating ecological security: Urbanization, spatial production and sustainable development

Rapid urbanization has caused a surge in water utilization, while climate change and human activity have reduced Qilian Mountains's ability to conserve water. Existing research shows

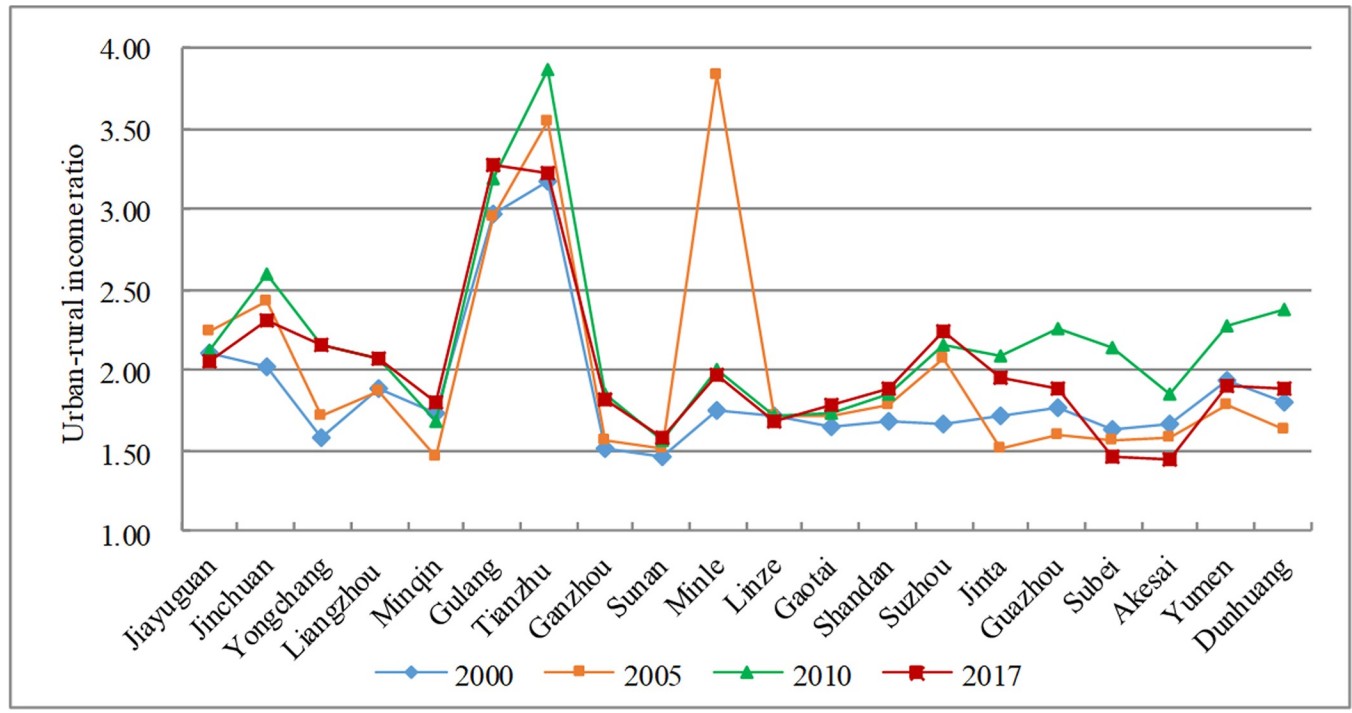

**Fig 8. Change of the urban-rural income proportion in the Hexi Corridor.**

**Table 1. Development and utilization of water resources in the Hexi Corridor.**

| Region | Surface water resources ($10^8$ m$^3$) | Unrepeated groundwater resources ($10^8$ m$^3$) | Water resources available ($10^8$ m$^3$) | Water consumption in 2017 ($10^8$ m$^3$) | Utilization ratio of water resources (%) | Water resources per capita (m$^3$/person) |
|---|---|---|---|---|---|---|
| Jiuquan Region | 20.26 | 7.02 | 27.28 | 27.28 | 100 | 2428 |
| Jiayuguan Region | 1.04 | 1.03 | 2.06 | 2.06 | 100 | 826 |
| Zhangye Region | 17.25 | 4.58 | 21.83 | 21.83 | 100 | 1776 |
| Wuwei Region | 4.38 | 2.34 | 6.72 | 6.72 | 100 | 1433 |
| JinchangRegion | 10.70 | 5.22 | 15.92 | 15.92 | 100 | 872 |
| Hexi Corridor | 53.63 | 20.19 | 73.82 | 73.82 | 100 | 1467 |

that when the water resource utilization reaches or exceeds its threshold [66], it will face a series of ecological problems such as increased desertification and oasis degradation, which will delay or even hinder social and economic development. The Hexi Corridor belongs to arid and water deficient regions in northwest China (Table 1). According to the classification criteria proposed by Fang Chuanglin et al [67]. Jiayuguan and Wuwei regions are the severely water deficient regions, and the water resources system very strongly resists the urbanization; Jinchang region is a moderate water deficient regions, and the water resource system is more resistant; Jiuquan region and Zhangye region are the water deficient regions, indicating that water resources are abundant and the resistance of the water resources system is relatively weak. Historically, water has been an important constraint to the Hexi Corridor's urbanization.

The land use/land cover area and the spatial structure of the built-up regions have undergone tremendous changes (Table 2), which is the spatial representation of the rapid development of urbanization [68]. From 2000 to 2015, the area of the built-up area of the Hexi Corridor increased by 368 km$^2$, and the average annual built-up area of each of the 20 cities in the region was 1.23 km$^2$. The growth rate is even faster than that of some large cities in developed coastal regions. According to the remote sensing data, the arable land area increased by 1452 km$^2$. But in fact, the urbanization takes up a lot of good arable land. In order to offset the impact of urbanization and maintain the red line of arable land set by the central government, the local government had to use other land resources for replacement, which reduced the bare land area by 1269 km$^2$. This unprecedented rate of urbanization has brought tremendous pressure on human-land coordination and sustainable development, which has severely damaged the local fragile ecological environment.

For the Hexi Corridor, the economic agglomeration and the concentration and migration of population caused by urbanization have adversely affected the use of land and water resources [69]. Inevitably, this pattern of urbanization caused the irreversible deterioration of the ecological environment, the over-exploitation of natural resources and the disappearance of biodiversity [6, 8, 28]. Taking the Qilian Mountain Nature Reserve as an example, due to the heavy development activities and the interference of human activities, the climate drought,

**Table 2. The area (km2) of land use changes during in 2000, 2005, 2010, and 2015.**

| Year | Cropland | Woodland | Grassland | Waterbody | Build-up land | Bare land |
|---|---|---|---|---|---|---|
| 2000 | 13324 | 6845 | 52662 | 2193 | 1061 | 167169 |
| 2005 | 14247 | 6811 | 52757 | 2235 | 1107 | 166818 |
| 2010 | 14341 | 6826 | 52770 | 2240 | 1114 | 166211 |
| 2015 | 14776 | 6817 | 52727 | 2327 | 1429 | 165900 |

the rising snow line, and grassland degradation have been affected to varying degrees, causing serious damage to the eco-environment. From the 1990s to the beginning of the 21st century, there has been local high-intensity development of hydropower projects in the basins of the Hei River, Shiyang River, and Shule River in the Qilian Mountains region, with more than 150 hydropower stations being constructed. Within the reserve alone, there are 532 large and small mining enterprises in Sunan, and 46 hydropower stations have been built on the main tributaries in Zhangye. As a result, more than 20,000 $m^2$ of the Qilian Mountain Reserve was occupied, and nearly 30,000 $m^2$ of vegetation was destroyed. In response to the ecological destruction of the Qilian Mountains, the relevant enterprises and people have been punished by the central government because of not being responsible for their duties. Seemingly, public administration forces represented by central government departments have curbed the momentum of ecological degradation.

But in fact, the real cause of the ecological and environmental problems in the Qilian Mountains is the spatial production process driven by multiple factors. This is reflected in how to regulate the government's behavior in the relationship between the environment and the economy. Then how to deal with the superior and subordinate power? How to use the relationship between power and capital to deal with the current environmental problems? How to use the power of the government to solve the problem of sustainable development? There is more than one such case in China, where illegal construction of new villas in the northern foothills of Qin Mountains and illegal reclamation of lake water in Qiandao Lake's drinking water reserve, the over-exploitation and destruction of vegetation in Muli Coalfield of Qinghai Province and the "contraction" of Karamaili Nature Reserve in Xinjiang reflect the encroachment of urbanization on resources and ecological environment. Therefore, coordinating and regulating the functions and interests of different roles in resource allocation and environmental protection in the urbanization is an important measure to achieve urban-rural integration and sustainable development.

## 4. Conclusions

The rapid urbanization has greatly changed the social and natural space of less-developed regions in China. However, in the Hexi Corridor area, the urbanization pattern dominated by spatial production has led some local governments to pursue digital urbanization and the blind expansion of the city scale. The excessive growth in fixed asset investment has severely curbed consumer demand. Real estate investment has been extremely inflated, forming the "ghost town". Over-exploitation of energy and excessive consumption of resources have caused serious damage to the ecological environment. The urban-rural income ratio in the region has been higher than the national average for a long time. The urban-rural gap in some regions has not narrowed, but is still widening [70]. These issues have shown that the spatial production driven by power and capital, while helping socio-economic development, ignores people-oriented social spaces that local people wanted, resulting in rural-urban separation. This made the contradiction between human and natural system is more acute. This spatial production pattern is also contradictory to sustainable development.

Urbanization of less-developed regions is a complex social process with significant spatial changes. Power plays a dominant role in the urbanization development of the Hexi Corridor. Although capital has participated in the spatial production, yet its impact on the urbanization of the Hexi Corridor is very limited due to the low-degree marketization. The spatial production has also caused serious ecological imbalances. This shows that the current urbanization pattern of the Hexi Corridor is unsustainable and must be changed [70]. So it is high time to rethink of traditional urbanization pattern and make a choice between spatial production and sustainable development.

Urbanization is a multi-dimensional social space process, and its occurrence and development are affected and restricted by many factors [49]. Different regions, because of different geographical, historical and socio-economic conditions, have different patterns of urbanization. The less-developed regions are highly resources dependent and fragile in ecological environment, so it is more difficult for them to achieve sustainable development than developed regions. Similarly, it is also important to improve the daily lives of residents, revitalize local cultural characteristics and protect the ecological environment [71]. Therefore, the urbanization pattern of less-developed regions should be diversified, which can be divided into three types: social type (power-oriented), economic type (capital-oriented) and environmental type (ecological-oriented). It is very urgent for the less-developed regions like Hexi Corridor to coordinate the relationship between urban and rural regions on different scales and transform the urbanization model from traditional spatial production to a new-type of urbanization with people-oriented and sustainable development [72].

## Supporting information

**S1 Data.**
(XLSX)

**S1 File. Data avalibility.**
(DOCX)

## Author Contributions

**Conceptualization:** Huailin Zhang, Zhibin Zhang.

**Data curation:** Huailin Zhang, Weimin Gong.

**Formal analysis:** Huailin Zhang, Jianhong Dong.

**Investigation:** Huailin Zhang, Fawen Gao, Wenbin Zhang.

**Methodology:** Huailin Zhang, Jianhong Dong, Weimin Gong.

**Resources:** Huailin Zhang, Fawen Gao, Wenbin Zhang.

**Writing – original draft:** Huailin Zhang.

**Writing – review & editing:** Huailin Zhang, Zhibin Zhang.

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
