## [Decision Letter · Decision Letter 0]

12 May 2020

PONE-D-20-09509

Spatial production or sustainable development?

Empirical research on the urbanization of less-developed regions based on the case of the Hexi Corridor in China

PLOS ONE

Dear Mr. Zhang,

Thank you for submitting your manuscript to PLOS ONE. After careful consideration, we feel that it has merit but does not fully meet PLOS ONE’s publication criteria as it currently stands. Therefore, we invite you to submit a revised version of the manuscript that addresses the points raised during the review process.

We would appreciate receiving your revised manuscript by Jun 26 2020 11:59PM. To enhance the reproducibility of your results, we recommend that if applicable you deposit your laboratory protocols in protocols.io, where a protocol can be assigned its own identifier (DOI) such that it can be cited independently in the future. For instructions see: http://journals.plos.org/plosone/s/submission-guidelines#loc-laboratory-protocols

We look forward to receiving your revised manuscript.

Kind regards,

Jun Yang

Academic Editor

PLOS ONE

Journal Requirements:

1. Please amend either the abstract on the online submission form (via Edit Submission) or the abstract in the manuscript so that they are identical.

2. We note you have included a table to which you do not refer in the text of your manuscript. Please ensure that you refer to Table 1 in your text; if accepted, production will need this reference to link the reader to the Table.

Additional Editor Comments (if provided):

Review 1

From temporal and spatial perspective, the article explored the evolution process of spatial production and urbanization, and pointed out that the urbanization mode of the Hexi Corridor must be changed. Some revisions need to be done below.

1) Line 149-153: “Based on the theories of sociology, economics, ecology and geography, the article proposes a new conceptual framework [44, 48, 53, 55, 56], it links spatial production, urbanization and the social space and ecological environment problems, and analyzes the process of spatial production and urbanization in less-developed.” Line 253-255: “It is not difficult to see that behind the investment has been marked by the power of the brand, but also further illustrates the influence of power on urbanization.” The sentences are very long and not clear, rephrase it.

2) Interdisciplinary research is currently more advocated by the academic community, and a variety of methods are used in the article to help solve and analyze problems. Please describe how the different analyses were conducted, and the contribution of each analysis to your findings.

3) Spatial production is the key term in the paper, however, it is difficult to understand the relationship between spatial production and sustainable development. Please add some details to make it clear.

4) The starting and ending time of this study is 2000-2017. But when analyzing the impact of urbanization on land use, remote sensing image data is provided in 2015. Please explain why.

5) Water shortage is an important factor affecting the ecological environment in Hexi Corridor. This article uses only 2017 data to analyze the use of water resources in the process of urbanization. Explain why.

6) Minor issues: I recommend the author(s) should find professional language agent to improve the English presentation style.

7) Last but not least, what is the intellectual merit and the novelty of the paper? What is the theoretical contribution of the paper? For example, what is new of this research compared with the studies of Fang et al., 2019; Ye et al., 2017; Tian and Sun, 2018. It is suggested that the author(s) think about these two issues carefully.

Review2

The topic of urbanization and sustainable development of less-developed regions in China is interesting and important. Especially, the paper explored the relationships between spatial production, urbanization and sustainable development based on the theory of spatial production, and found that the mode of urbanization in Hexi Corridor is unsustainable and must be transformed. However, there are some minor problems in this paper.

1. The framework of the authors put is novel. Describe in detail the literature form which theories and frameworks are drawn.

2. In the case study part, despite the authors present the materials about the irreversible deterioration of the ecological environment, the authors are lack of criticism of their argument to help international readers to better understand the current Chinese urbanization.

3. If it is possible, I suggest the authors provide one more paragraph to illustrate the structure of this paper at the end of section 1.

4. Is the term "social space dialectics" expressed accurately? Please pay attention to the specifications expressed in English. There are some terminology errors, for example, “space production” and “stratum”, must be carefully revised.

5. There some problems with sentence structure, verb tense, and clause construction. All these problems should be improved.

Reviewers' comments:

Reviewer's Responses to Questions

**Comments to the Author**

1. Is the manuscript technically sound, and do the data support the conclusions?

Reviewer #1: Yes

Reviewer #2: Yes

2. Has the statistical analysis been performed appropriately and rigorously? 

Reviewer #1: Yes

Reviewer #2: Yes

3. Have the authors made all data underlying the findings in their manuscript fully available?

Reviewer #1: Yes

Reviewer #2: Yes

4. Is the manuscript presented in an intelligible fashion and written in standard English?

Reviewer #1: Yes

Reviewer #2: No

5. Review Comments to the Author

Reviewer #1: From temporal and spatial perspective, the article explored the evolution process of spatial production and urbanization, and pointed out that the urbanization mode of the Hexi Corridor must be changed. Some revisions need to be done below.

1) Line 149-153: “Based on the theories of sociology, economics, ecology and geography, the article proposes a new conceptual framework [44, 48, 53, 55, 56], it links spatial production, urbanization and the social space and ecological environment problems, and analyzes the process of spatial production and urbanization in less-developed.” Line 253-255: “It is not difficult to see that behind the investment has been marked by the power of the brand, but also further illustrates the influence of power on urbanization.” The sentences are very long and not clear, rephrase it.

2) Interdisciplinary research is currently more advocated by the academic community, and a variety of methods are used in the article to help solve and analyze problems. Please describe how the different analyses were conducted, and the contribution of each analysis to your findings.

3) Spatial production is the key term in the paper, however, it is difficult to understand the relationship between spatial production and sustainable development. Please add some details to make it clear.

4) The starting and ending time of this study is 2000-2017. But when analyzing the impact of urbanization on land use, remote sensing image data is provided in 2015. Please explain why.

5) Water shortage is an important factor affecting the ecological environment in Hexi Corridor. This article uses only 2017 data to analyze the use of water resources in the process of urbanization. Explain why.

6) Minor issues: I recommend the author(s) should find professional language agent to improve the English presentation style.

7) Last but not least, what is the intellectual merit and the novelty of the paper? What is the theoretical contribution of the paper? For example, what is new of this research compared with the studies of Fang et al., 2019; Ye et al., 2017; Tian and Sun, 2018. It is suggested that the author(s) think about these two issues carefully.

Reviewer #2: The topic of urbanization and sustainable development of less-developed regions in China is interesting and important. Especially, the paper explored the relationships between spatial production, urbanization and sustainable development based on the theory of spatial production, and found that the mode of urbanization in Hexi Corridor is unsustainable and must be transformed. However, there are some minor problems in this paper.

1. The framework of the authors put is novel. Describe in detail the literature form which theories and frameworks are drawn.

2. In the case study part, despite the authors present the materials about the irreversible deterioration of the ecological environment, the authors are lack of criticism of their argument to help international readers to better understand the current Chinese urbanization.

3. If it is possible, I suggest the authors provide one more paragraph to illustrate the structure of this paper at the end of section 1.

4. Is the term "social space dialectics" expressed accurately? Please pay attention to the specifications expressed in English. There are some terminology errors, for example, “space production” and “stratum”, must be carefully revised.

5. There some problems with sentence structure, verb tense, and clause construction. All these problems should be improved.

6. PLOS authors have the option to publish the peer review history of their article (what does this mean?). If published, this will include your full peer review and any attached files.

Reviewer #1: None

Reviewer #2: No

---

## [Author Response · Author response to Decision Letter 0]

4 Jun 2020

Dear editor and reviewers:

Thank you very much for your suggestions. We would like to submit the revised manuscript entitled “Spatial production or sustainable development? An empirical research on the urbanization of less-developed regions based on the case of Hexi Corridor in China” (PONE-D-20-09509), which we wish to be considered for publication in your Journal. Those comments are all valuable and very helpful for revising and improving our paper. We have studied the comments carefully and made corrections which we hope meet with approval. The main corrections in the paper and responding to the reviewer’s comments are as follows.

Reviewer 1:

1) Line 149-153: “Based on the theories of sociology, economics, ecology and geography, the article proposes a new conceptual framework [44, 48, 53, 55, 56], it links spatial production, urbanization and the social space and ecological environment problems, and analyzes the process of spatial production and urbanization in less-developed.” Line 253-255: “It is not difficult to see that behind the investment has been marked by the power of the brand, but also further illustrates the influence of power on urbanization.” The sentences are very long and not clear, rephrase it.

ANS: We have rephrased these sentences.

Based on the theories of sociology, economics, ecology and geography, the article proposes a new conceptual framework [44, 48, 53, 55, 56], links spatial production, urbanization and the social space and ecological environment problems, and analyzes the process of spatial production and urbanization in less-developed regions.

However, the change in investment structure reflects the role of the government, which further illustrates the impact of power on urbanization.

2) Interdisciplinary research is currently more advocated by the academic community, and a variety of methods are used in the article to help solve and analyze problems. Please describe how the different analyses were conducted, and the contribution of each analysis to your findings.

ANS: In order to analyze the relationship between spatial production and urbanization, we used Ye Chao and Yang Zhen’s research methods to construct a simple indicator system. In the process of explaining the evolution of urbanization and space production in Hexi corridor, this paper analyzes it from two dimensions of time and space. In terms of time, it combs the country and local policy of the same period. Using the statistical data, the paper clarified the main driving force and role of urbanization in underdeveloped areas under multiple factors. In terms of space, we used the GIS method to visualize the spatial production process, and analyzed the spatial process and dynamics of the urbanization of the Hexi Corridor from the county level. We mainly illustrate the impact of urbanization or spatial production on sustainable development from two aspects: water resources and land resources. The paper uses the RS method to interpret land use resource data, and local government data (from the Water Resources Bulletin) to explain the dynamics and characteristics of changes.

3) Spatial production is the key term in the paper, however, it is difficult to understand the relationship between spatial production and sustainable development. Please add some details to make it clear.

ANS: As a critical social or urban theory, “production of space generally means that the urban landscapes and spatial structures have been reshaped by political, economic, and social factors, mainly including capital, power, and class, so that the urban space finally becomes their production and process” (Ye et al., 2014). This concept extends the core idea of production of space, “(social) space is (social) production” (Lefebvre, 1991), and here “social” is a concept in a broad sense, including political, economic, (narrow-sense) social and cultural and other factors or human behaviors. Spatial production is not only the reproduction of social space, but also the reproduction of spatial relationships. Space is the product and content of society, and society is the collection and representation of space. The two are constantly being reshaped under the interweaving action (Ye et al., 2014). Space is not just a passive recipient of social, economic and cultural infestation, but an active participant in urbanization (Lv et al., 2019). Capital, power, and class are the sources of motivation for spatial production and social revolution. Spatial production is a social relationship under the influence of various forces like capital, and then this social relationship has shaped an urbanization process from rural space to urban space (Fang et al., 2015). These processes directly affect the three sustainable development areas of economic prosperity, social development and environmental protection (Claudia and Karola, 2019).

4) The starting and ending time of this study is 2000-2017. But when analyzing the impact of urbanization on land use, remote sensing image data is provided in 2015. Please explain why.

ANS: The LULC data of this paper are taken from the Resource and Environment Data Cloud Platform (http://www.resdc.cn/) of Chinese Academy of Sciences. The data is based on Landsat TM/ETM+/OLI remote sensing Image, which is generated by manual visual interpretation, and the synthetic precision is over 95%. Currently, land use data for eight periods in 1980, 1990, 1995, 2000, 2005, 2010, 2015 and 2018 are available. Since there is no 2017 land use data, the 2015 data are used instead in this paper.

5) Water shortage is an important factor affecting the ecological environment in Hexi Corridor. This article uses only 2017 data to analyze the use of water resources in the process of urbanization. Explain why.

ANS: Rapid urbanization of Hexi Corridor faced a series of ecological problems such as increased desertification and oasis degradation, which will delay or even hinder social and economic development. The Hexi Corridor belongs to arid and water deficient regions in northwest China. We collected water resource data of Hexi Corridor in 2000,2005,2010 and 2017, which are from the annual “Water Resources Bulletin” issued by the Gansu Provincial Department of Water Resources. According to the international standards (UNDP, 1990), the Hexi Corridor is divided into four water deficient regions. From 2000 to 2017, we found that water resource only in Wuwei region went from moderate to severe water shortage, other areas did not change significantly. To avoid duplication, we only use the water data of 2017 to reflect the water use situation in the Hexi Corridor.

6) Minor issues: I recommend the author(s) should find professional language agent to improve the English presentation style.

ANS: We have asked the professional language agent (see attachment) and professional staff to modify the English expression. All of corrections in text have been marked in color.

7) Last but not least, what is the intellectual merit and the novelty of the paper? What is the theoretical contribution of the paper? For example, what is new of this research compared with the studies of Fang et al., 2019; Ye et al., 2017; Tian and Sun, 2018. It is suggested that the author(s) think about these two issues carefully.

ANS: Spatial production theory is concerned with the relationship between society and space. According to social-spatial dialectics, political power, economic capital and social class are the three important components of spatial production (Ye et al., 2017). The traditional pattern of urbanization, dominated by government power, with large-scale expansion of space and capital, is almost at odds with the goal of sustainable development (Ye et al., 2017), this social relationship reshapes the process of urbanization from rural space to urban space. However, spatial production-led urbanization is changing human-land relationships as well as socio-cultural spaces in less developed regions (Fu et al., 2018). Urban-rural disparities have increased, spatial injustices have become more frequent, resource overuse and ecological damage have seriously affected regional sustainable development (Fang et al., 2019). The harmonious development of population, economy, resources and environment is the foundation of sustainable development (Tian and Sun, 2018). It is an inevitable choice for sustainable development to study the process and mechanism of spatial production and urbanization, and to explore the urbanization mode in harmony with ecological environment. On the basis of the above-mentioned theories, this paper puts forward a new, conceptual framework by using the existing theoretical framework for reference. Previous researches mainly concentrate on spatial production in developed countries or regions. The urbanization and sustainable development of less-developed regions, with complex and fragile ecological environments, are often overlooked. It is a new idea to explain the relationships and interactions between spatial production, urbanization and sustainable development based on less-developed regions by the theory of spatial production. This paper proposes a new framework of “spatial production-urbanization-sustainable development”, which helps to formulate policies and meet challenges in less-developed regions, and provides a scientific reference for sustainable urbanization in developing countries.

Reviewer 2:

1. The framework of the authors put is novel. Describe in detail the literature form which theories and frameworks are drawn.

ANS: According to Lefebvre's social-spatial dialectics, political power, economic capital and social class are the three important components of space production (Ye et al., 2017). Spatial production theory is concerned with the relationship between society and space. The traditional pattern of urbanization, dominated by government power, with large-scale expansion of space and capital, is almost at odds with the goal of sustainable development (Ye et al., 2017), this social relationship reshapes the process of urbanization from rural space to urban space. However, spatial production-led urbanization is changing human-land relationships as well as socio-cultural spaces in less developed regions (Fu et al., 2018). The gap between urban and rural areas is widening, space inequality is becoming more pronounced, and the excessive use of resources and ecological damage are seriously affecting sustainable development in different regions (Fang et al., 2019). The harmonious development of population, economy, resources and environment is the basis of sustainable development (Tian and Sun, 2018). It is the inevitable choice of sustainable development to study the process and mechanism of space production and urbanization, and to explore the mode of urbanization coordinated with the ecological environment. On the basis of the above-mentioned theories, this paper puts forward a new conceptual framework by using the existing theoretical framework for reference.

2. In the case study part, despite the authors present the materials about the irreversible deterioration of the ecological environment, the authors are lack of criticism of their argument to help international readers to better understand the current Chinese urbanization.

ANS: We have commented on the case in the last paragraph of section 3. But in fact, the real cause of the ecological and environmental problems in the Qilian Mountains is the spatial production process driven by multiple factors. This is reflected in how to regulate the government’s behavior in the relationship between the environment and the economy. Then how to deal with the superior and subordinate power? How to use the relationship between power and capital to deal with the current environmental problems? How to use the power of the government to solve the problem of sustainable development? Therefore, coordinating and regulating the functions and interests of different roles in resource allocation and environmental protection in the process of urbanization is an important measure to achieve urban-rural integration and inclusive development.

3. If it is possible, I suggest the authors provide one more paragraph to illustrate the structure of this paper at the end of section 1.

ANS: We have added a paragraph to illustrate the structure of this paper at the end of section 1 according to the reviewer’s suggestion.

4. Is the term “social space dialectics” expressed accurately? Please pay attention to the specifications expressed in English. There are some terminology errors, for example, “space production” and “stratum”, must be carefully revised.

ANS: “social space dialectics” should be corrected to “social-spatial dialectics”, “space production” should be “spatial production”, and “stratum” should be “class”. All of these corrections in text have been marked in color.

5. There some problems with sentence structure, verb tense, and clause construction. All these problems should be improved.

ANS: We have asked professional language agents (see attachment) and professionals to help us modify the inappropriate English expressions in the article. All corrections in the text have been coloured in.

We have tried our best to improve the manuscript and made some changes in the manuscript. These changes will not influence the content and framework of the paper. We appreciate for Editors/Reviewer’ warm work earnestly, and hope that the corrections will meet with approval. We would like to express our heartfelt thanks to the editors/reviewers for their enthusiastic work, and it is hoped that these corrections will be approved.

Thank you very much again for your comments and suggestions.

Best

Authors

---

## [Decision Letter · Decision Letter 1]

15 Jun 2020

Spatial production or sustainable development? An empirical research on the urbanization of less-developed regions based on the case of Hexi Corridor in China

PONE-D-20-09509R1

Dear Dr. Zhang,

We’re pleased to inform you that your manuscript has been judged scientifically suitable for publication and will be formally accepted for publication once it meets all outstanding technical requirements.

Kind regards,

Jun Yang

Academic Editor

PLOS ONE

Additional Editor Comments (optional):

Accept

Reviewers' comments:

Reviewer's Responses to Questions

**Comments to the Author**

1. If the authors have adequately addressed your comments raised in a previous round of review and you feel that this manuscript is now acceptable for publication, you may indicate that here to bypass the “Comments to the Author” section, enter your conflict of interest statement in the “Confidential to Editor” section, and submit your "Accept" recommendation.

Reviewer #1: All comments have been addressed

Reviewer #2: All comments have been addressed

2. Is the manuscript technically sound, and do the data support the conclusions?

Reviewer #1: Yes

Reviewer #2: Yes

3. Has the statistical analysis been performed appropriately and rigorously? 

Reviewer #1: Yes

Reviewer #2: Yes

4. Have the authors made all data underlying the findings in their manuscript fully available?

Reviewer #1: Yes

Reviewer #2: Yes

5. Is the manuscript presented in an intelligible fashion and written in standard English?

Reviewer #1: Yes

Reviewer #2: Yes

6. Review Comments to the Author

Reviewer #1: (No Response)

Reviewer #2: (No Response)

7. PLOS authors have the option to publish the peer review history of their article (what does this mean?). If published, this will include your full peer review and any attached files.

Reviewer #1: No

Reviewer #2: No

---

## [Editor Report · Acceptance letter]

25 Jun 2020

PONE-D-20-09509R1 

Spatial production or sustainable development? An empirical research on the urbanization of less-developed regions based on the case of Hexi Corridor in China 

Dear Dr. Zhang:

I'm pleased to inform you that your manuscript has been deemed suitable for publication in PLOS ONE. Congratulations! Your manuscript is now with our production department. 

Kind regards, 

on behalf of

Dr. Jun Yang 

Academic Editor

PLOS ONE